# Influence of Block-Copolymers’ Composition as Compatibilizers for Epoxy/Silicone Blends

**DOI:** 10.3390/molecules28176300

**Published:** 2023-08-28

**Authors:** Christelle Delaite, Sophie Bistac, Daniela Rusu

**Affiliations:** Laboratoire de Photochimie et d’Ingenierie Macromoleculaires (LPIM EA 4567), Université de Haute-Alsace, F-68100 Mulhouse, France; sophie.bistac-brogly@uha.fr (S.B.); daniela.rusu@mines-paris.org (D.R.)

**Keywords:** epoxy network, PDMS, compatibilizers, polymers blends

## Abstract

The objective of this study was to prepare crosslinked epoxy networks containing liquid silicone particles in order to improve their mechanical properties and obtain less brittle materials. Different copolymers were used as compatibilizers. These copolymers vary in their chemical composition and structure. All of the copolymers contain hydrophobic (PDMS sequences) and hydrophilic groups. The effect of their chemical structure and architecture on the morphology of the dispersed phase, and on the final physico-chemical and flexural characteristics of epoxy/silicone blends, was explored. The morphology of crosslinked formulations was studied by scanning electron microscopy (SEM), and the thermal characteristics (glass transition temperature, T_g_, and curing exothermic peak) were determined by differential scanning calorimetry (DSC). The experimental results have shown that the average diameter and particle size distribution of silicone particles depend on the chemical structure and architecture of the compatibilizers. One copolymer has been identified as the best compatibilizer, allowing a lower mean diameter and particle size distribution in addition to the best mechanical properties of the final network (less brittle character). This study has consequently evidenced the possibility of creating in situ silicone capsules inside an epoxy network by adding tailored compatibilizers to epoxy/silicone formulations.

## 1. Introduction

Epoxy resins form a class of widely used high-performance thermosetting polymers, due to their excellent mechanical properties, moisture and chemical resistances, heat and dimensional stabilities, very good electrical insulation, and very good adhesion to different substrates [1,2,3]. These engineering performances and the large versatility of epoxy resin formulations have designated them as perfect thermosetting resins for a broad range of applications, from coatings and electronics encapsulation to structural applications and adhesives. Usually, the epoxy thermosetting formulations and curing conditions are chosen based upon the performance requirements for the end-product [4,5]. However, fully-cured epoxy thermosets have an inherent low toughness and impact resistance, because of their high modulus and glass transition temperatures induced by their highly crosslinked 3D structure [1,2,3,4,5,6,7]. Thus, toughening epoxy thermosets has become one of the most investigated topics by industrial and academic communities.

Several strategies have been developed to consistently increase the epoxy toughness without a significant lowering of the overall mechanical performance or the glass transition temperature (T_g_). One of the first approaches has been to add reactive liquid elastomers such as carboxyl-terminated butadiene-acrylonitrile copolymer (CTBN rubber) to the uncured epoxy formulations and to cure these reactive systems [1,2,7,8,9]. During epoxy crosslinking, phase separation occurred and generated biphasic microstructures with small rubber particles homogeneously dispersed inside the cured epoxy matrix [1,7,9]. The mechanisms of this strategy of rubber-toughening in epoxy networks involve the capacity of rubber particles to stop crack formation and propagation through high-crosslinked epoxy networks by: (i) rubber bridging the two etches of the crack, (ii) local shear bands that may occur between rubber particles, and (iii) the possible cavitation of rubber particles, which is able to stop the propagation of crack sharp tips [8,10,11,12,13,14].

A more recent version of the rubber-toughening approach proposes ready-to-use, well-defined core-shell rubbery particles instead of reactive liquid elastomers. These core-shell micro-/nanoparticles typically present a rubbery core (i.e., polybutadiene, polysiloxane) and a shell able to ensure strong bonding to the resin matrix. Since this strategy generates similar biphasic final morphologies as in the case of reactive liquid rubbers, it was assumed that the mechanisms described for rubber-toughening are also valid here [1,7,15,16,17,18,19]. However, two advantages of using well-defined core-shell elastomeric particles were pointed out, namely: (i) the relative control of the viscosity of epoxy formulations (easier to process) and (ii) the better control of the nature of particles/epoxy matrix interfaces [1,7,20].

A very recent strategy to epoxy-toughening suggests using self-organizing thermoplastic copolymers that are miscible with the uncured epoxy matrix, and conduct phase-separation during epoxy curing, generating different nanostructured elastomeric domains inside the 3D epoxy network. The major advantage of this approach consists of the much lower viscosity of the epoxy formulation, making it easier to process [1,18,19,20,21,22,23].

Other recent exploratory strategies for epoxy-toughening propose using nanoparticles such as silica [24,25,26], graphene [27,28], TiO_2_ [27,29], and carbon nanotubes [30,31,32,33]. It has even been shown that silica nanoparticles can be as effective as rubber particles in improving the fracture toughness/energy of epoxy composites [20,34,35,36]. However, synergetic mixtures [20,37,38] of reactive liquid elastomers and nanoparticles were reported as more effective to cumulate the toughening effects of both rubber micro-/nanodomains and nanoparticles. The involved mechanisms depend on the nature of nanoparticles and epoxy systems [1,20,24,39].

In addition to the interest in improving epoxy toughness and impact resistance, some certain new challenges have appeared in recent years, based on the great potential of epoxy resins to produce innovative advanced functional materials exhibiting self-healing, self-cleaning, shape memory, or other functional properties such as transparent-to-opaque transitions [1,28,40,41,42,43,44].

In the present work, we investigated the possibility of preparing 3D epoxy networks containing liquid silicone particles and tailoring their biphasic morphology by choosing an appropriate compatibilizer [45]. Using a chemically inert polydimethylsiloxane (PDMS) of relatively low molar mass allowed for the prospecting of the possible migration through the epoxy matrix network, as has been reported for rigid thermoplastics [46], and understanding of the role it could play in toughening epoxy thermosets. Considering that silicone and epoxy resin are immiscible in the range of temperatures under study (25 to 150 °C), five different compatibilizers were selected to stabilize the morphology of epoxy/silicone blends. The effect of their chemical structure and architecture on the morphology of the dispersed phase, and on the final physico-chemical and mechanical characteristics of epoxy/silicone blends, was explored.

## 2. Results and Discussion

### 2.1. Determination of the EEW of DGEBA by ^1^H NMR 

The equivalent epoxy weight (EEW) of a DGEBA resin represents very important data for developing formulations with a stoichiometric epoxy/hardener ratio. For the present work, the DGEBA supplier’s data only indicated a batch with ${\bar{\;M}}_{n}$ < 700 g·mol^−1^. Consequently, the exact EEW and ${\bar{\;M}}_{n}$ values were determined based on the experimental determination of the DGEBA average polymerization degree, ${\bar{DP}}_{n}$, via ^1^H NMR data [47,48], knowing that:$${\bar{\;M}}_{n}=M_{\mathrm{DGEBA}\;\mathrm{repetitive}\;\mathrm{unit}}*{\bar{DP}}_{n}+M_{\mathrm{DGEBA}\;(n=0)}<=>\;{\bar{M}}_{n}=284*{\bar{DP}}_{n}+340$$
which gives:$$EEW_{DGEBA}=142*\;{\bar{DP}}_{n}+170$$

On the other hand, ${\bar{DP}}_{n}$ can be expressed as a function of the proton intensities from DGEBA ^1^H NMR spectrum, as follows [47,48]:$${\bar{DP}}_{n}=(R_{p}-R_{t})/R_{t}$$
where *R_p_* is the ratio between the proton signals associated with the aromatic rings of DGEBA repetitive units, (*I_a+b_*) and the proton signals corresponding to the terminal epoxy units (*I_g+h+e_*), and *R_t_* corresponds to the same ratio in the case of the DGEBA oligomer (when *n* = 0).

Figure 1 presents the ^1^H NMR spectrum of DGEBA used in this study.

The results show that the DGEBA resin used in this study has a ${\bar{DP}}_{n}$ of 0.12 and an EEW of $187\;g\cdot {\mathrm{eq}}^{-1}$. The experimental exact value of ${\bar{\;M}}_{n}$ is 374 g·mol^−1^, well below the 700 g·mol^−1^ indicated by the supplier. Consequently, for all formulations, the experimental EEW value was taken for the DGEBA/IPD ratio.

### 2.2. Miscibility Tests of Compatibilizers

The results of the miscibility tests are presented in Table 1 (see Appendix A). The miscibility was assumed as reasonable (M) when a binary mixture was completely transparent and remained transparent during five hours. Binary mixtures forming white instable emulsions were considered as only partially miscible (PM) or immiscible (IMM).

The last column from Table 1 indicates the partners for the premixing stage: the copolymers with long EO and/or PO segments and terminal hydroxyl groups were premixed with DGEBA resin, while the copolymers with long silicone segments and terminal-epoxy groups were premixed with the PDMS, i.e., with the future minor phase of the polymer blends.

### 2.3. Curing of Epoxy/Silicone Resins

Preliminary DSC measurements have shown that the neat DGEBA/IPD epoxy system reached full crosslinking after one hour of curing at 150 °C (see Appendix A). Based on these data, the following final curing protocol was adopted for all epoxy formulations: mixing at 33 °C during 10 min in order to fluidify the DGEBA/IPD mixture and to obtain a homogeneous mixture, then a temperature ramp up to 150 °C, and a post-curing stage at 150 °C for one hour. At the beginning of the pre-curing stage, all epoxy/silicone thermosets behaved as immiscible polymer blends in a viscous liquid state, with homogeneous dispersed PDMS inclusions inside the continuous DGEBA + IPD matrix. The biphasic morphologies corresponding to the same set of mixing conditions were stabilized by using silicone–polyether copolymers with different chemical structures and architectures.

The main chemical reactions for DGEBA + IPD resins are presented in Figure 2. The high reactivity of epoxy groups is due to their stressed three-membered ring structure, and it allows them to react with a wide variety of organic compounds with active hydrogen atoms [1,3,4,5,6,7]. On the other hand, the amine groups of IPD are known to present different reactivities, depending on their position in the cycle [49,50].

Figure 2a–c also show that crosslinking reactions generate hydroxyl groups in the system, which are known to accelerate the formation of a fully crosslinked epoxy network [6,7,49]. In addition, in cases where the amine concentration is locally low or not in a stoichiometric ratio to epoxy, or in cases where the secondary amine is less reactive than the hydroxyl groups from the system, epoxy groups may react with these hydroxyl groups (Figure 2d) [1,3].

During the curing of all of the epoxy systems, the formation of the first crosslinks induced an increase in the matrix viscosity and glass temperature transition. The curing phase at a slow temperature ramp (2.5 °C·min^−1^) offered additional heat for epoxy and diamine to regain thermodynamic mobility and continue the crosslinking reactions. The post-curing stage for one hour at 150 °C was intended to fully crosslink the epoxy network.

The curing behavior of epoxy reactive systems (with and without silicone) was observed by non-isothermal DSC, from room temperature to 210 °C, at 10 °C·min^−1^. The onset of cure (T_onset_), the exothermic peak temperature (T_peak_), and the heat of cure ΔH_cure_ of the epoxy formulations were determined from the DSC curves.

Figure 3 presents a typical DSC heating scan of uncured epoxy formulations. Full crosslinking was associated with the return of the DSC heat flow to a quasi-linear response.

The presence of liquid inclusions of 5 to 10 wt% of chemically inert silicone did not change the large exothermic profile due to the curing of the epoxy matrix (see Appendix A). When comparing the curing parameters for neat epoxy resins and in the presence of 5 to 10 wt% silicone, no significant effect could be observed for T_onset_ or the temperature of the maximum curing rate, T_peak_. For instance, Table 2 presents the curing data corresponding to neat epoxy and, respectively, epoxy formulations with 5 to 10% silicone and coPo3 as a compatibilizer (1 wt% with respect to the silicone amount).

The only difference concerns the decrease in ΔH_cure_ with the addition of silicone that corresponds to the decrease in epoxy matrix concentration. Similar results have been obtained for the formulations containing the other compatibilizers.

### 2.4. Morphological Analysis

The morphological analysis of cured systems was performed by SEM. Cured neat epoxy resins presented a homogeneous transparent aspect. At the end of the curing procedure, all epoxy/silicone formulations containing PDMS and the compatibilizers presented biphasic morphologies with well-dispersed spherical particles inside a continuous rigid epoxy network. Table 3 presents the morphological characteristics of the epoxy/silicone blends, with and without compatibilizer.

The SEM images show that all of the compatibilizers assured a good epoxy/silicone compatibilization, especially for 5 wt% silicone (see Figure 4). All investigated epoxy blends with 5 wt% and 10 wt% PDMS presented spherical silicone liquid inclusions, regardless the compatibilizer type (Figure 4 and Figure 5).

At equal concentrations (1 wt% copolymer, with respect to the silicone content), the most efficient compatibilizers for epoxy/silicone resins based on DGEBA are coPo1 and 3, which gave fine morphologies with narrow particle size distributions. Both copolymers present linear hydrophilic polyether chains which are OH terminated.

### 2.5. Thermal and Dynamic Mechanical Characteristics of Cured Epoxy Formulations

Table 4 presents the glass transition temperature of cured epoxy formulations, as obtained by DSC and DMA measurements. Figure 6 and Appendix A present typical DMA and DSC curves obtained for the cured neat DGEBA/IPD epoxy and the cured epoxy/silicone (10% of PDMS) with coPo3, respectively. With the exception of some differences between the T_g_ values obtained by DSC and DMA, which are not unusual since these two methods are not describing the same physical phenomena from the sample, no clear dependence could be detected depending on the particle sizes of the silicone inclusions or the effectiveness of the different compatibilizers.

However, DMA data allowed further evaluation of the effect of silicone and compatibilizers on the thermomechanical characteristics and crosslink density (ρ_crosslink_) of epoxy/silicone blends. According to the theory of rubber elasticity, the storage modulus G’ from the rubbery plateau (generally taken at T_g_ + 30 °C, at the beginning of the rubbery plateau) is directly proportional to the crosslink density of cured resins but can also be influenced by the presence of PDMS inclusions [51,52,53,54].

Figure 6 shows the typical evolution of the storage modulus and mechanical loss factor and the influence of adding inert liquid silicone in epoxy formulations.

A gradual drop in the storage modulus with the addition of 5 and 10 wt% inert silicone, respectively, was globally observed, which normally reflects an increase in the flexibility of epoxy formulations. Furthermore, this gradual drop was also observed for the values of G′ from the rubbery plateau.

### 2.6. Contact Angle Measurements

Contact angle measurements were performed to investigate the potential migration of the liquid silicone chains or compatibilizer molecules at the cured epoxy surface from the inclusions. Measurements carried out 10 days and 1 month, respectively, after curing the epoxy/silicone blends gave stable water wettability results, within the method precision (Table 5 and Figure 7).

The contact angle of pure DGEBA/IPD resin is in agreement with values from other works [55,56]. Cured epoxy/silicone formulations with 5% PDMS content, compatibilized with copolymers having long hydrophilic polyether arms oriented to the epoxy matrix (coPo1, coPo3), presented more hydrophilic surfaces than pure epoxy resin. These copolymers were the most efficient as compatibilizers (see Table 4).

On the opposite side, the use of coPo2, coPo4, and coPo5 induced a more hydrophobic surface which is probably linked to their lower efficiency to disperse PDMS inside the epoxy resin. Thus, a part of the PDMS chains can also migrate at the surface and generate a certain enrichment of these surfaces in PDMS, with respect to the bulk concentration, explaining the higher hydrophobicity.

Considering that a lower hydrophobicity of the cured resin is required for painting or bonding to assure a good wetting and adherence between the cured epoxy and another surface (metal substrate, coating or adhesive applications, etc.), this study suggests that the most appropriate compatibilizers for reaching this goal are coPo1 and coPo3.

### 2.7. Flexural and Fracture Behavior

Figure 8 presents the influence of silicone inclusions and different compatibilizers on the flexural behavior of cured epoxy formulations. Typically, adding liquid silicone inclusions reduced the flexural moduli of cured resins, in accordance with the DMA results. This effect is more visible for the blend containing 10% PDMS.

Since coPo1 and coPo3 exhibit the best compatibilizing capacities, their mechanical properties will be compared. An increase in the flexural strength (at break) is observed for the blend containing coPo1, for both PDMS contents, contrary to the blend containing coPo3 for which a decrease in the flexural strength can be noticed. For the other formulations, no significant trend is observed, proving that the mechanical properties depend not only on the dispersion quality but also on other parameters such as the chemical architecture (linear chains or with a lateral group) or the chemical composition (presence of a reactive group like an epoxy function. Finally, the deflection at break increased globally for all samples compared to the neat epoxy. However, higher values were measured for the blends containing coPo1.

Fracture micrographs of different blends show spherical inclusions inside the cured epoxy matrix, which have a liquid silicone core able to flow during fracture (coPo5, Figure 9). Without significantly modifying the flexural modulus or strength, the best compatibilizers (coPo1 and coPo3) could improve the deflection at break (Figure 8c).

The experimental results show that all copolymers allow a stable dispersion of liquid silicone inclusions inside the epoxy network (without significantly modifying the curing reaction and the final T_g_). However, some of them prove to be more effective. In particular, coPo1 and coPo3 induce smaller particle sizes and a narrower distribution, for both of the PDMS concentrations studied. These two linear copolymers with polyether/silicone molar ratios close to one and OH terminal groups are consequently the most efficient as compatibilizers. The ether and hydroxyl groups allow a good compatibility with DGEBA chains, which also possess OH and ether groups, while their linear PDMS sequences allow a good compatibility with the PDMS dispersed phase.

In addition, the formulations with coPo1 and coPo3 present a higher hydrophilicity (lower contact angle of water), which proves that both PDMS and copolymers macromolecules are well included in the network, avoiding any migration at the surface. This property is important for coating or bonding epoxy surfaces.

The study of the flexural properties, in particular the deformation at break, indicates an improvement (less brittle character) for all of the copolymers, which was the objective of this study. Nevertheless, coPo1 makes it possible to obtain the most interesting properties, with a higher deformation at break.

## 3. Materials and Methods

### 3.1. Materials

The high-purity diglycidyl ether of bisphenol-A (DGEBA, RENLAM CY219) was purchased from Huntsman Advanced Materials, Europe, Basel, Switzerland. The curing agent, isophorone diamine (IPD), was purchased from Sigma-Aldrich, Saint-Quentin-Falavier, France. Trimethylsiloxy-terminated polydimethylsiloxane (PDMS, M_w_ = 28,000 g·mol^−1^) was purchased from Alfa Aesar, Illkirch, France. The different linear di-functional or multi-pendant silicone polyethers used as compatibilizers (Figure 10 and Table 6) were kindly provided by Siltech Co., Toronto, ON, Canada. All components are liquids and were used as received.

### 3.2. Preparation and Curing of Epoxy Resin Formulations

Preliminary miscibility tests of the different silicone–polyethers copolymers with DGEBA and PDMS defined the order of mixing. Depending on their thermodynamic affinity, the copolymers were premixed with either DGEBA or PDMS for 5 min, at 33 °C, in a silicone bath. Then, PDMS or DGEBA was added and mechanically mixed. The ratio between the silicone–polyether copolymers and non-reactive PDMS was kept constant at 1 wt% for all formulations. Finally, stoichiometric amounts of IPD were added (DGEBA/IPD weight ratio = 4.4) and mixed for another 10 min at 33 °C. Thermosetting formulations were poured into different silicone molds to obtain rectangular specimens for physico-chemical and mechanical characterizations. The formulations were pre-cured in an oven at 33 °C for 60 min, then heated with a temperature ramp up to 150 °C at 2.5 °C·min^−1^, and then post-cured for 60 min at 150 °C.

### 3.3. Characterization Methods

#### 3.3.1. ^1^H NMR Analysis for Determining the Epoxy Equivalent Weight of DGEBA

^1^H NMR analyses were performed with a Bruker 300 UltraShield spectrometer, Bruker, Marne La Vallée, France, at 300 MHz, driven by the XWIN-NMR V3.5 software. DGEBA epoxy resin was dissolved in deuterated chloroform at a concentration of 10 wt%.

#### 3.3.2. Miscibility Tests

Miscibility tests of the different compatibilizers with either DGEBA or silicone were performed at room temperature and 33 °C (in a heating bath), to determine the best pre-mixing procedure for optimally incorporating all the components of the epoxy formulations. Equal quantities of silicone–polyether copolymers and either DGEBA or PDMS were vigorously mixed during 10 min, and the visual aspect of the mixture was recorded for five hours.

#### 3.3.3. Morphology Analysis

Morphological analysis of epoxy formulations was performed by scanning electron microscopy (SEM). The morphology of fractured specimens of cured resins was also investigated by SEM, by means of a JEOL JCM 6000, Jeol, Croissy-sur-Seine, France. The fractured specimens of cured resins were previously immersed in butyl acetate (a good solvent for PDMS) for 24 h, at room temperature, to remove the silicone contamination from the surface and potentially from a few microns in depth. After removal from butyl acetate, the samples were dried and analyzed by SEM. The particle size distribution and average diameter value of inclusions were determined by image analysis, after verifying the homogeneity of the morphological patterns over 3 samples for each specimen.

#### 3.3.4. Differential Scanning Calorimetry (DSC)

DSC was used to investigate the curing of epoxy formulations, to determine the T_g_, and to study the influence of silicone inclusions on the curing kinetics of epoxy materials. The DSC analyses were carried out with a TA Instruments’ DSC-Q200, TA Instruments, Guyancourt, France under a nitrogen atmosphere, using hermetically sealed aluminum pans.

A first heating scan was applied on uncured and cured samples, from 25 to 200 °C, followed by cooling down to 25 °C, and a second heating scan from 25 to 200 °C; all scans were performed at constant rate of 10 °C·min^−1^. In the case of cured samples, the first heating scan was used to verify whether the curing was complete or not, and the second to determine the T_g_ of the cured samples. The first heating scan applied on uncured epoxy formulations indicated the onset temperature of curing, and the exothermic peak temperature. The area under the exothermic peak due to the crosslinking reaction gave, by integration, the heat of cure, ΔH_cure_ (J/g). The second heating scan was used as previously, to determine the T_g_ of in situ cured formulations.

#### 3.3.5. Dynamic–Mechanical Analysis (DMA)

Dynamic–mechanical analyses were performed on a modular Anton Parr MCR 302 rheometer, Anton Paar, Les Ulis, France. Cured rectangular specimens (40 mm × 12 mm × 2.3 mm) were mounted in single cantilever bending clamps and submitted to a temperature scan from 25 to 180 °C, with a heating rate of 2 °C·min^−1^. The viscoelastic moduli, storage (G′) and loss modulus (G″), and mechanical loss tangent (tan *δ*) were obtained for a frequency of 1 Hz. The glass-transition temperature values were assumed as the temperatures at the maximum tan *δ* peak.

#### 3.3.6. Flexural Properties

The flexural properties of cured epoxy formulations were measured using a Testometric M500-30AT universal testing machine, Testometric, Rochdale, UK. Three-point bending tests were performed on rectangular specimens (142 mm × 12 mm × 2.3 mm). The measurements were performed with a crosshead speed of 0.5 mm/min, at room temperature. Seven specimens were tested for each epoxy formulation.

#### 3.3.7. Contact Angle

Contact angle measurements were performed to evaluate the water wettability of cured epoxy resins with and without silicone in the formulation. The experiments were carried out with a Krüss DSA100 Drop Shape Analyzer, Krüss, Villebon sur Yvette, France using the sessile drop method. Water droplets of 0.2 µL were deposited on epoxy surfaces (both upper and lower surfaces of rectangular specimens), with a 0.53 mm diameter syringe, at a 500 µL/min flow rate. At least 10 measurements were conducted for each surface.

## 4. Conclusions

This study has evidenced the possibility of creating in situ liquid silicone capsules inside an epoxy network and to control their size and distribution by adding compatibilizers to epoxy/silicone formulations.

The experimental results have shown that the average diameter and particle size distribution of silicone capsules depend on the chemical structure and architecture of the silicone-compatibilizers. The copolymers coPo1 and coPo3 were the most efficient as compatibilizers. CoPo1 appears to be the best choice to decrease the brittle character of epoxy networks while improving the hydrophilicity of the cured sample. The next step will be to study aging by following the evolution of properties over time and under aggressive conditions (temperature, humidity, etc.).

## Figures and Tables

**Figure 1 molecules-28-06300-f001:**
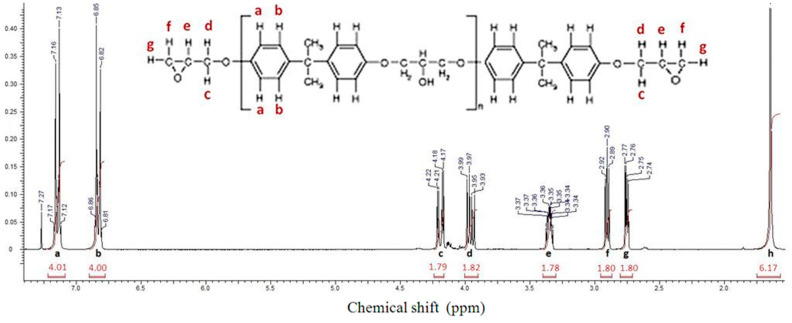
^1^H NMR spectrum of DGEBA RENLAM CY219.

**Figure 2 molecules-28-06300-f002:**
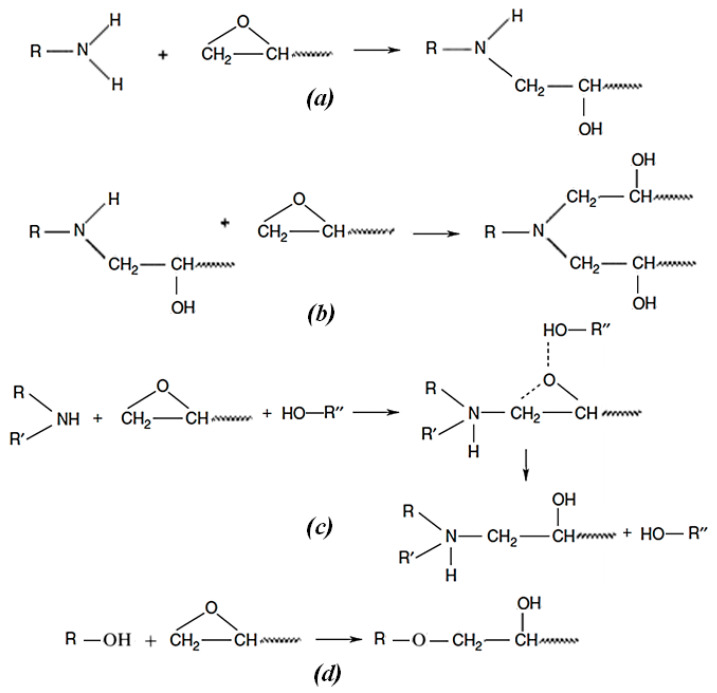
Chemical reactions during crosslinking of DGEBA resin with a diamine [1,3,4,5,6,7,49,50].

**Figure 3 molecules-28-06300-f003:**
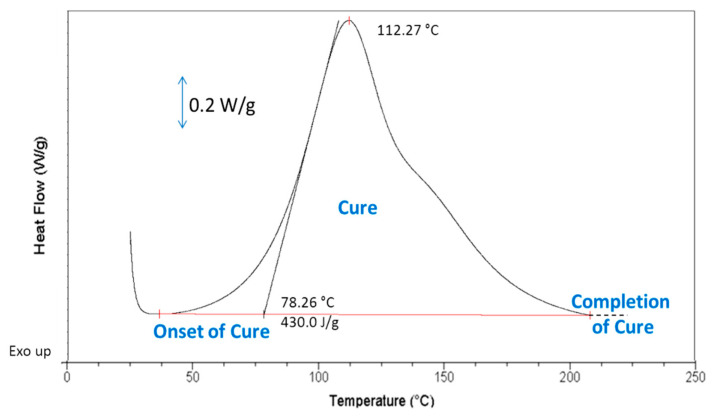
DSC heating scan of uncured neat epoxy resin (first ramp).

**Figure 4 molecules-28-06300-f004:**
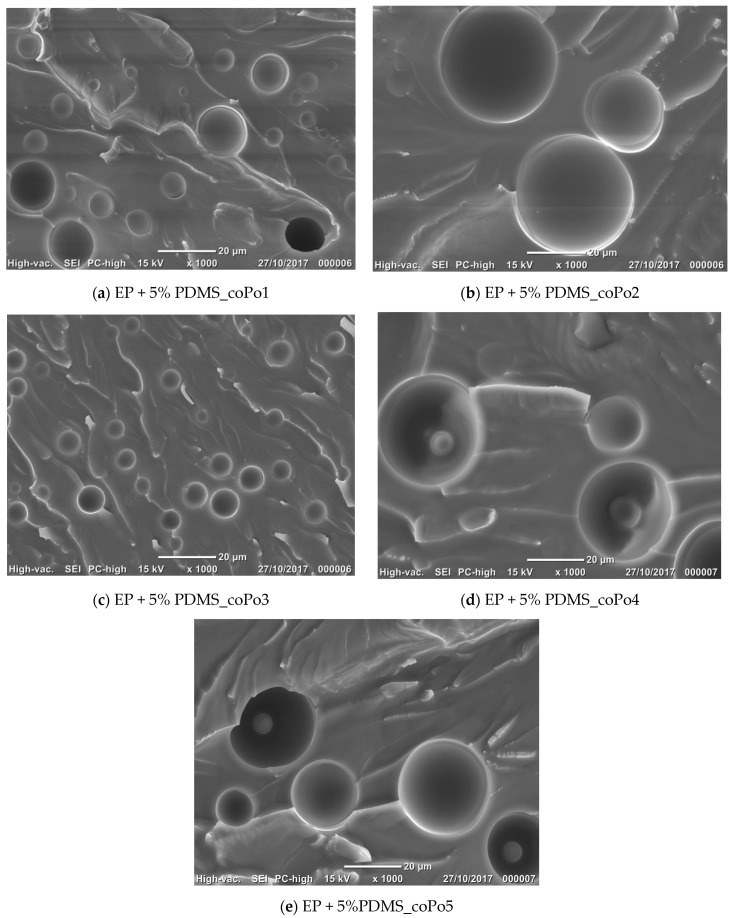
Scanning electron micrographs of fracture surfaces of the epoxy/silicone blends (5% PDMS) and different compatibilizers.

**Figure 5 molecules-28-06300-f005:**
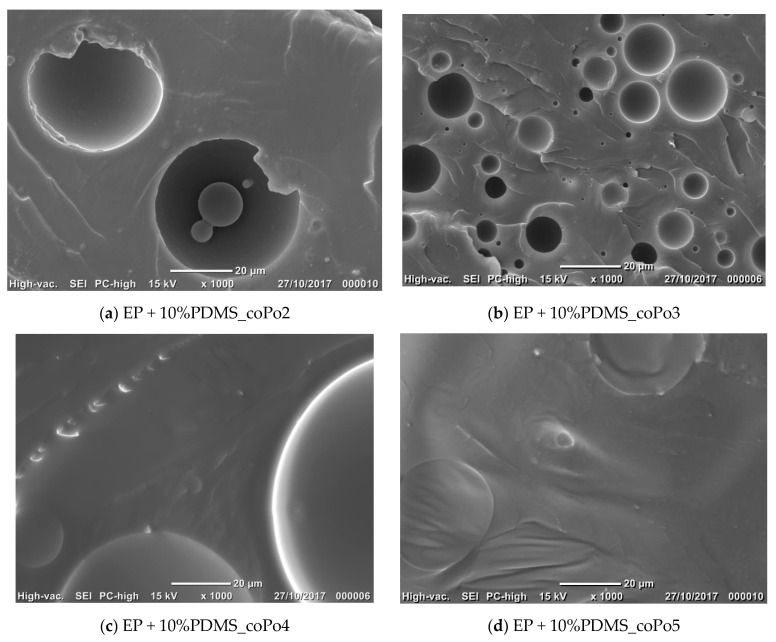
Scanning electron micrographs of fracture surfaces of the epoxy/silicone blends (10% PDMS) compatibilized with coPo2, coPo3, coPo4, and coPo5, respectively.

**Figure 6 molecules-28-06300-f006:**
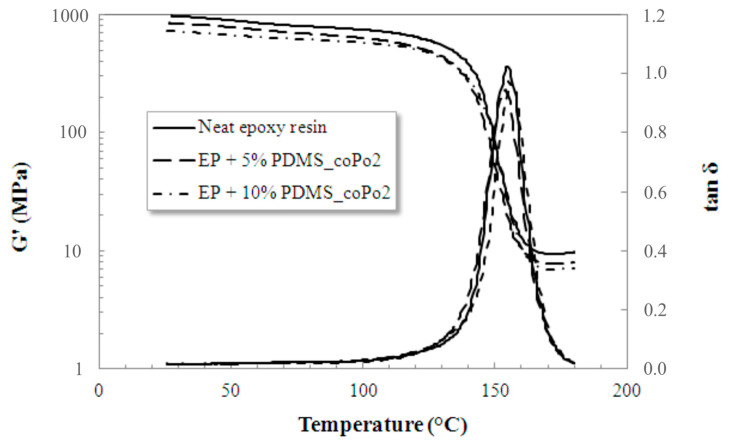
Dynamic mechanical behavior of the pure epoxy network vs. epoxy/silicone blends compatibilized with coPo2.

**Figure 7 molecules-28-06300-f007:**

Contact angles of water drops on different surfaces after 10 days: (**a**) Neat epoxy resin; (**b**) Epoxy formulation with 5% PDMS and coPo1; (**c**) Epoxy formulation with 5% PDMS and coPo5.

**Figure 8 molecules-28-06300-f008:**
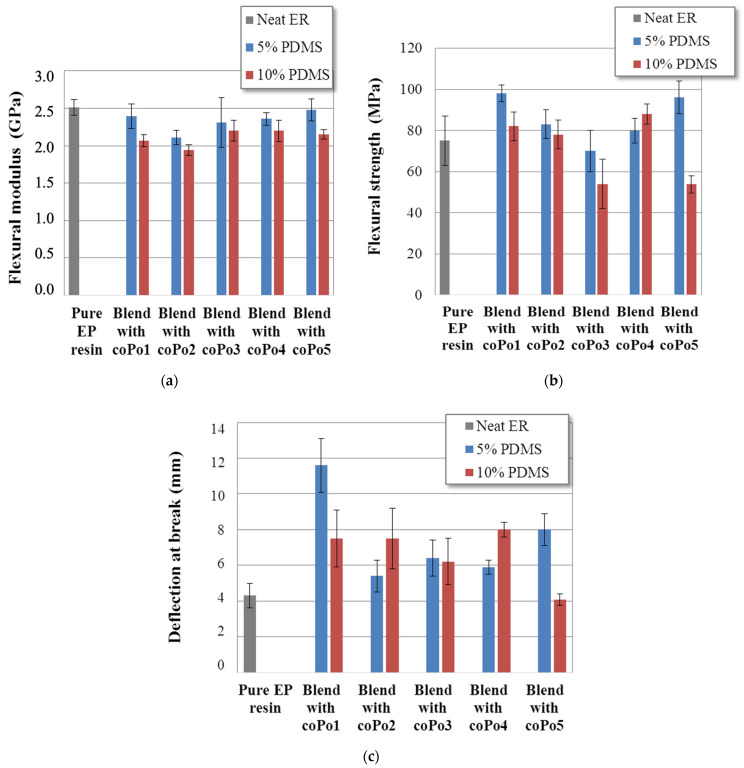
Flexural characteristics of pure epoxy resin and epoxy/silicone blends with different compatibilizers: (**a**) flexural modulus, (**b**) flexural strength, and (**c**) deflection at break.

**Figure 9 molecules-28-06300-f009:**
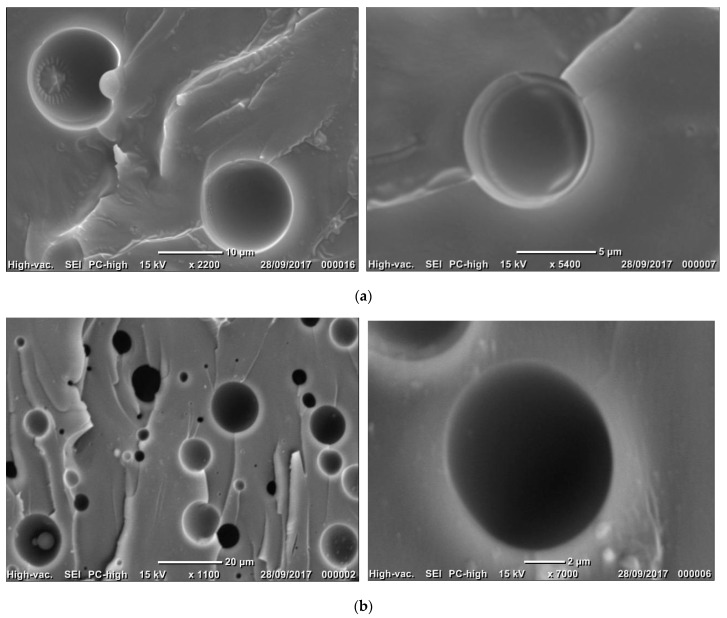
SEM images of the fracture of formulations with 5% PDMS and (**a**) coPo3 or (**b**) coPo5, respectively. Right side—higher magnification.

**Figure 10 molecules-28-06300-f010:**
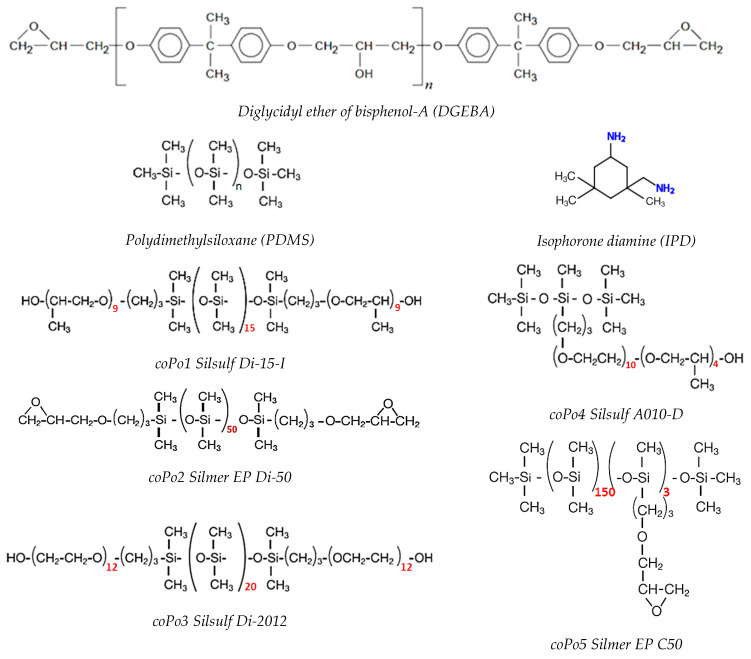
Chemical structures of the different components of epoxy/silicone formulations.

**Table 1 molecules-28-06300-t001:** Results of the miscibility tests.

Copolymer Code	Miscibility at 25 °C and 33 °C		Order of Addition in the Thermosetting Formulation
	DGEBA	PDMS	
coPo1	PM	PM/IMM	Pre-mixed with DGEBA
coPo2	PM/IMM	M	Pre-mixed with PDMS
coPo3	M	PM/IMM	Pre-mixed with DGEBA
coPo4	M	PM/IMM	Pre-mixed with DGEBA
coPo5	PM/IMM	M	Pre-mixed with PDMS

**Table 2 molecules-28-06300-t002:** Curing characteristics obtained by DSC.

Formulation Code	ΔH_cure_ (J·g^−1^)	T_onset_ (°C)	T_peak_ (°C)
Pure epoxy resin (EP)	430	78.0	112.3
EP + 5% PDMS and coPo3	405	76.5	111.7
EP + 10% PDMS and coPo3	383	75.7	112.1

**Table 3 molecules-28-06300-t003:** Morphological characteristics of silicone spherical capsules and size distribution.

Formulations		Morphological Characteristics	
		Mean Diameter (µm)	Size Distribution
Uncompatibilized epoxy/silicone blends			Broad
coPo1	5% PDMS	8.6	Narrow
	10% PDMS	11.7	Narrow
coPo2	5% PDMS	9.0	Narrow
	10% PDMS	29.4	Broad
coPo3	5% PDMS	5.2	Narrow
	10% PDMS	7.8	Narrow
coPo4	5% PDMS	28.3	Broad
	10% PDMS	48.1	Broad
coPo5	5% PDMS	8.4	Narrow
	10% PDMS	26.5	Broad

**Table 4 molecules-28-06300-t004:** Glass transition temperatures (by DSC and DMA) and storage modulus of the different cured epoxy formulations.

Formulations		T_g_ (°C) by DSC	T_g_ (°C) by DMA	G’_rubbery @_ (T_g_ + 30 °C) (MPa)
Pure epoxy resin		155	155	9.8
coPo1	5% PDMS	152	154	7.7
	10% PDMS	148	153	6.5
coPo2	5% PDMS	153	154	7.9
	10% PDMS	151	155	7.1
coPo3	5% PDMS	155	158	10
	10% PDMS	153	154	9.8
coPo4	5% PDMS	150	159	9.4
	10% PDMS	148	155	7.1
coPo5	5% PDMS	151	156	7.6
	10% PDMS	150	157	4.7

**Table 5 molecules-28-06300-t005:** Contact angle values of water droplets on epoxy formulations after 10 days.

Contact Angle (°)			
Pure DGEBA/IPD epoxy resin		80 ± 2	
Epoxy formulations with 5 wt% PDMS		Epoxy formulations with 10 wt% PDMS	
coPo1	74 ± 2	coPo1	83 ± 2
coPo2	86 ± 5	coPo2	95 ± 2
coPo3	77 ± 3	coPo3	85 ± 2
coPo4	81 ± 3	coPo4	87 ± 3
coPo5	86 ± 3	coPo5	92 ± 3

**Table 6 molecules-28-06300-t006:** Main physico-chemical characteristics of epoxy resin formulations’ components.

Components	Molar Mass * (g·mol^−1^)	Specificities/Equivalent Weight * (g·eq^−1^)	Functionality Type/Number	Possible Reaction with
DGEBA	${\bar{\;M}}_{n}$ < 700	Not indicated	Epoxy/F = 2	-NH_2_; -OH
IPD	M = 170	AHEW = 42.58	-NH_2_/F = 4	epoxy groups
PDMS	M_w_ = 28,000	trimethylsiloxy terminated	-/F = 0	-
coPo1	M_w_ = 2400	PO/silicone molar ratio = 18/15	-OH/F = 2	epoxy groups
coPo2	M_w_ = 4100	EEW = 2050 long silicone chain	Epoxy/F = 2	-NH_2_; -OH
coPo3	M_w_ = 2900	EO/silicone molar ratio = 24/20	-OH/F = 2	epoxy groups
coPo4	M_w_ = 900	EO/PO/silicone molar ratio = 10/4/1	-OH/F = 1	epoxy groups
coPo5	M_w_ = 11,800	EEW = 3900 long PDMS chain	Epoxy/F = 3	-NH_2_; -OH

* Suppliers’ data. PO = propylene oxide unit. EO = ethylene oxide unit. EEW = equivalent epoxy weight. AHEW = amine hydrogen equivalent weight.

## Data Availability

Not applicable.

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
