# Peer review of "Influence of Block-Copolymers’ Composition as Compatibilizers for Epoxy/Silicone Blends"

_molecules, 2023, doi:10.3390/molecules28176300_

Round 1

Reviewer 1 Report

Comments to the Author

This research prepared crosslinked epoxy networks containing liquid silicone particles to improve their mechanical properties and obtain less brittle materials. Different copolymers were used as compatibilizers. The effect of their chemical structure and architecture on the morphology of the dispersed phase, and on the final physico-chemical and flexural characteristics of silicone-epoxy blends was explored. Major revision is needed and some of them are listed as follows:

Q1: Line 30-34, “These engineering performances and the large versatility of epoxy resin formulations have designated them as perfect thermosetting resins for a broad range of applications, from coatings and electronics encapsulation, to structural applications and adhesives. Usually, the epoxy thermosetting formulations and curing conditions are chosen upon the performance requirements for the end product” which seem to lack references. Check for similar issues in the full text.

Q2: Line 115, Figure 1 is a blurred image, please replace it.

Q3: Line 91, the EEW recommends acetone-hydrochloric titration for determination.

Q4: Line 124, The results of the miscibility tests should provide relevant pictures.

Q5: Line 144, “Preliminary DSC measurements in isothermal mode (not shown here) have shown that neat DGEBA/IPD system reached full crosslinking after one hour of curing at 150°C.” Please provide evidence of full crosslinking, such as the complete disappearance of epoxy peaks observed by in-situ infrared spectroscopy.

Q6: Line 147 “a curing stage for one hour at 33 °C”, but as shown in Figure 3, the curing heat release of epoxy resin at 33 °C is not obvious. What is the significance of choosing 33 °C curing for 1 hour?

Q7: Line 154-169 does not explain the reaction process and mechanism during epoxy resin crosslinking in Figure 2c.

Q8: Line 185, “The presence of liquid inclusions of 5 to 10 wt% of chemically inert silicone did not change the large exothermic profile due to the curing of epoxy matrix.” Please provide the relevant DSC heating scan.

Q9: In Figure 4e, the ruler of Scanning electron micrographs is 10 microns, for comparing particle size distributions, it is recommended to use the same ruler as the other images in Figure 4.

Q10: Line 231, “Table 4 presents the glass transition temperature of cured epoxy formulations, as obtained by DSC and DMA measurements.” Please provide the relevant pictures curve of DSC and DMA.

Q11: Line 260-263, “A gradual drop of storage modulus with the addition of 5 and respectively 10 wt% inert silicone was observed, which normally reflects an increase in flexibility of epoxy formulations. Furthermore, this gradual drop was also observed for the values of G’ from the rubbery plateau.” But in Table 4, the G’ value of Copo3 5% PDMS is 13.2, which is higher than 9.8 of pure epoxy resin, the trend contradicts that of other samples. Please explain.

Q12: For coPo4 10% PDMS and coPo5 5% PDMS in Figure 8, which are not the best compatibilizers, their flexural strength and the deflection at break are better than coPo3 that the best compatibilizers, please explain this difference.

Author Response

Dear reviewer 1,

thanks for your comments on our article entitled "Influence of Block Copolymers Composition as Compatibilizers for Epoxy-Silicone blends".

We hope that the answers and additional information provided in the attached file will meet your expectations.

Withe my best regards,

Prof C. Delaite

Reviewer 2 Report

The author demonstrate in the manuscript the influence of PDMS in epoxy system. I would recommend publication of this manuscript after major revision.

1) Epoxy blend are well know system and has adverse effect due to moisture. In presence of PDMS does the system exhibit any decrease in the properties due to moisture? Was the system tested in different humidity level?

2) Was variation was observed in the stiffness of the material upon increase the level of PDMS ?

3) What is Elongation at break and elastic modulus of these material? How did the % variation of PDMS / copolymers?

4) Does the material has any dampening effect?

5) Was any fatique test conducted? 

6) What is the gel point of all the system? Any diffusion limitation was observed during the curing process. 

Author Response

Dear reviewer 2,

thank you for your comments on our article entitled "Influence of block-copolymers composition as compatibilizers for epoxy-silicone blends".

We hope that the answers  provided in the attached file will meet your expectations.

With my best regards,

Prof C. Delaite
